# Wool Keratin Nanofibers for Bioinspired and Sustainable Use in Biomedical Field

**DOI:** 10.3390/jfb14010005

**Published:** 2022-12-21

**Authors:** Diego Omar Sanchez Ramirez, Claudia Vineis, Iriczalli Cruz-Maya, Cinzia Tonetti, Vincenzo Guarino, Alessio Varesano

**Affiliations:** 1CNR-STIIMA (National Research Council-Institute of Intelligent Industrial Technologies and Systems for Advanced Manufacturing), Corso Giuseppe Pella 16, 13900 Biella, Italy; 2CNR-IPCB (National Research Council-Institute for Polymers, Composites and Biomaterials), Mostra, d’Oltremare, Pad. 20, V.le J.F. Kennedy 54, 80125 Napoli, Italy

**Keywords:** wool keratin, electrospinning, nanofibers, biocompatible, biodegradable

## Abstract

Keratin is a biocompatible and biodegradable protein as the main component of wool and animal hair fibers. Keratin-based materials support fibroblasts and osteoblasts growth. Keratin has been extracted by sulphitolysis, a green method (no harmful chemicals) with a yield of 38–45%. Keratin has been processed into nanofibers from its solutions by electrospinning. Electrospinning is a versatile and easy-to-use technique to generate nanofibers. It is an eco-friendly and economical method for the production of randomly and uniaxially oriented polymeric nanofibers. Thanks to their high specific surface area, nanofibers have great potential in the biomedical field. Keratin nanofibers have received significant attention in biomedical applications, such as tissue engineering and cell growth scaffolds, for their biocompatibility and bio-functionality. Accordingly, we propose an extensive overview of recent studies focused on the optimization of keratinbased nanofibers, emphasizing their peculiar functions for cell interactions and the role of additive phases in blends or composite systems to particularize them as a function of specific applications (i.e., antibacterial).

## 1. Introduction

Wool is a natural fiber mainly made of fibrous proteins—keratin. The increased interest in using biomass and natural raw materials attracts the use of this protein [1] largely present in different sites from human and animal sources (i.e., nails, hair, wool feather horn and hooves [2]). Differently to the other fibrous proteins, keratin is characterized by its high content of sulfur groups due to the presence of cysteine-rich proteins that form the characteristic fibrous structure through intra- and inter-molecular disulfide bonds providing the good mechanical properties and stability of keratin-based materials (Figure 1) [3]. Moreover, the molecular weight and sulfur content of keratin can vary throughout a fiber. As a structural protein, keratin has the key role of supporting the structural shape of wool fibers by a wide range of inter-/intra-molecular interactions (i.e., hydrogen bonds, electrostatic/hydrophobic/hydrophilic and disulfide bonds with effects on the spatial configuration of chains (i.e., α−helix or β−sheet), Figure 1. Among them, disulfide bonds are directly responsible for the chemical stability in aqueous solutions preventing the dissolution of fibers in water and biological media. Hence, it is explained the strong interest of using keratin as a biomaterial with good biocompatibility and biodegradability suitable to produce scaffolds for biomedical applications (i.e., wound healing and hemostasis) [4]. Recent studies have demonstrated the ability of wool keratin to support the adhesion and growth of different cell phenotypes including fibroblasts [5], osteoblasts [6], neuroblasts and keratinocytes [1,2]. This is strictly related to the presence of cellular binding motifs such as arginine–glycine–aspartic acid (RGD), glutamic acid-aspartic acid-serine (EDS) and leucine–aspartic acid–valine (LVD) that promote the cell attachment, so reproducing the interaction patterns supported by native Extra Cellular Matrix-like proteins [7,8].

In order to exploit the natural properties of wool keratin (WK) as a structural protein, after extraction and purification, WK can be processed to produce scaffolds as interconnected structures similar to Extra Cellular Matrix (ECM). This goal can be reached by electrospinning, a technique that generates nanofibrous systems that mimic the structural environment of ECM and support the growth of cells in applications such as tissue engineering [9,10] and dental implants [5,11]. In wound dressing, WK-nanofibers could also have applications in producing nanofibers membranes with antibacterial properties [12,13] and drug delivery systems [14,15]. The typical applications of WK-nanofibers have been presented in Figure 2. Finally, it is essential to remember that the electrospinning process can be extended to many other polymers (i.e., collagen, chitosan, polycaprolactone and polyhydroxy butyrate [16,17,18]) to obtain nanofibers with drug delivery systems [19] and antimicrobial properties [20] for therapeutic use [21] and wound healing applications [22].

It is worth mentioning that the research on nanofibers (NFs) production by electrospinning has been growing worldwide over time [23,24]. According to the amount of publications in Scopus and Web of Science, China, United States, South Korea, Iran, India, Germany, Japan, United Kingdom, Turkey and Italy currently report the major number of publications in this field. Among them, Italy and China have published the most significant number of articles on the production and application of WK-NFs. As part of the special issue “State-of-the-Art Functional Biomaterials in Italy”, this review will deal with scientific publications that have been exclusively conducted in Italy.

WK is not only a bioinspired protein capable of supporting cell growth, but also it is a sustainable biopolymer, which can be extracted from the woolen by-products of the textile industry and recycled wool clothing. Indeed, only in the United States at least 16 × 10^6^ tons of used textile waste are produced per year according to the U.S. Environmental Protection Agency [25]. As a result, WK can be extracted from discarded animal hair fibers by sulphitolysis and, successively, purified in order to obtain aqueous solutions of pure WK. Afterward, this protein is lyophilized and its powder can be dissolved in water, non-aqueous solvents and mixed with other polymers to produce WK-NFs, as reported in Table 1. The typical non-aqueous solvents used in the electrospinning process of WK are formic acid (FA) and hexafluoroisopropanol (HFIP).

In the literature, WK has been electrospun pure and in blend with different kinds of polymer (synthetic, natural, biodegradable/non-biodegradable and water/non-water soluble), as seen in Table 1. The addition of other polymers allows transferring the inherent properties of those to the WK-biobased scaffold, for example, mechanical properties. On one hand, polyethylene oxide (PEO), polyvinylpyrrolidone (PVP) and polyamide 6 (PA6) are the most common non-biodegradable polymers electrospun with WK that improve the electrospinnability of this protein and keep the biocompatibility of NFs. On the other hand, polybutylene succinate (PBS), polycaprolatone (PCL), polylactic acid (PLA), fibroin (FIB), gelatin (GEL), sericin (SER) and polyvinyl alcohol (PVA) are the most popular biodegradable and biocompatible polymers employed in blend with WK to produce nanofibrous membranes. To summarize, Figure 3 reports the different solutions employed during the electrospinning of WK, which has been sorted out according to the solvent (aqueous/non-aqueous) and the biodegradability of polymers.

Table 2 shows the multiple electrospinning conditions (solvent, polymer concentration, voltage, flow rate and so on) that have to be employed in the production of WK-NFs. As evidenced by Figure 3 and Table 2, WK is a versatile protein that can be electrospun under different conditions in water, non-aqueous solvents and offers a wide range of options for polymer blends. Despite the fact that all those polymers are biocompatible, not all of them are biodegradable. Therefore, this review will discuss the morphology, the stability of NFs in water, the selection of solvent, the response of WK-NFs during in vitro test and the potential application of these functional biomaterials, considering mainly three groups: pure WK-NFs, NFs made of non-biodegradable polymers with WK and NFs made of biodegradable polymers with WK.

## 2. Pure WK—NFs

In the last two decades, WK has been processed in different forms—i.e., films, hydrogels, sponges and nanofibers [40,41,42,43]. In particular, electrospun NFs are preferred due to their high surface area to volume and microporous structure that allows supporting cell adhesion, proliferation and migration [44,45].

Indeed, micro- and nano-structural properties play a relevant role on the local interface with cells, mediated by the spatial organization of macromolecular chains of keratin and chemical functionalities of WK suitable for cells recognition. In this context, the structural properties can also be influenced by the protein interactions with solvents in solution. After extraction, pure WK-NFs can be generally obtained by using FA and HFIP. These two solvents tend to impart different micro and nano-structural properties which may significantly alter the local microenvironment and its interface with cells [10]. In the case of FA, the fiber morphology can be more accurately controlled, also showing a more homogeneous size distribution compared to those prepared with HFIP that, instead, promote the formation of several defects along the fiber body—i.e., beads, flat ribbons (Figure 4) [10].

The key role of fiber morphology is also confirmed by in vitro studies that indicated a higher proliferation of cells after 14 days in the presence of WK-NFs fibers without defects and more controlled spatial organization. According to previous studies reported in the literature, pure keratin electrospun NFs were preferentially obtained by dissolving the protein amounts in formic acid from 15 to 20 wt.%. The keratin solution at higher concentration tends to produce NFs with higher mean diameter, whereas very thin and very homogeneous nanofibers were produced from less concentrated solutions (mean diameter of 169 nm and standard deviations of 49 nm) although a major number of beads was recognized [10].

Starting from these experiments, highly oriented WK-NFs were recently proposed to increase the gingival fibroblast adhesion and proliferation on titanium surfaces to prevent peri-implantitis and epithelial down-growth after the dental implant treatment [5]. Results showed a strong influence of WK-NFs on fibroblast proliferation with respect to the grooves of the titanium surfaces.

Independently upon the keratin source (i.e., hair, feathers), keratin-based materials also present a well-recognized antimicrobial activity that can be enhanced by the incorporation of a wide variety of antimicrobial agents and ion compounds. (i.e., silver) [9]. In this context, morphological cues (i.e., fiber diameter and preferential alignment) can corroborate the biological response, supporting the growth of fibroblasts. For instance, a recent work suggested the use of aligned fibers to support the regeneration of periodontal tissue around the collar of trans-mucosal dental implants. In this case, the preferential orientation of WK-NFs in a circumferential way allows stimulating fibroblast interactions (Figure 5), addressing new tissue growth along the titanium surface plug [11].

One of the main drawbacks in the fabrication of biologically stable substrates is related to the conditions used during the extraction process, that may drastically influence the degradation mechanisms of the protein. This involves the chemical or enzymatic attack of the disulfide groups, deputed to determine chain rigidity and protein solubility in aqueous environment [46]. In this context, it is well known that disulfide bridges can support the hydrogen bonds’ action, thus promoting more intense interactions among keratin chains. Hence, a strict control of these processes may ensure the reproduction of α helix-like structure that is typical of other biological proteins such as collagen, also improving stiffness and mechanical strength [47].

In this view, recent studies based on the use of hydrolyzed keratins have demonstrated a strong interest to use NFs for biomedical applications, making available tripeptides sequences Arg–Gly–Asp (RGD) and Leu–Asp–Val (LDV) able to link the most cell surface ligands in order to promote cell–cell or cell–matrix interactions required to support adhesion and cell proliferation mechanisms. For instance, Fortunato et al. proposed the use of hydrolyzed-keratin-based NFs with controlled chemical/physical features in terms of swelling, contact angle, mechanical properties and surface charge density to develop bioinspired substrates to be used as in vitro tissue models [48]. They demonstrated that hydrolyzed keratins can be successfully used to fabricate innovative biomaterials to study cell interactions to collect more information on the peculiar behavior of selected cell phenotypes into an ECM-simulated microenvironment.

Noteworthy, technological approaches mainly oriented to improve the biomechanical stability of keratins involve the reuse of keratins derived from by-products of animal industry, such as wool fibers, animal hair and feathers. However, in order to fully preserve biocompatible properties, a successful approach concerns processing them in combination with other biopolymers (often other structural proteins and polysaccharides) that have been demonstrated to be more suitable for the fabrication of bulk devices (i.e., sponges, film, scaffolds) to be used for different applications from tissue engineering to the drug delivery [34,49]. In the next section, further examples of blended fibers including WK will be discussed by underlining their potential applications in the biomedical field.

## 3. WK and Non-Biodegradable Polymers—NFs

Non-biodegradable biocompatible polymers are used in bio-stable implanted devices as structural systems in the body. In particular, polyamides have been found to be safe for use as scaffolds and nanofillers to improve the mechanical properties of composite materials [50]. In a research work [27], WK has been blended with PA6 at different ratios highlighting the immiscibility of the polymers with a viscometric method and observing segregation phases in films, Figure 6a. Despite this finding, the NFs obtained by the electrospinning of the blends had good homogeneity and no phase segregations because of the rapid solvent evaporation occurring in electrospinning, Figure 6b. The NFs morphology was investigated according to the electrospinning process parameters by principal component analysis, revealing that the percentage of keratin is negatively correlated to increasing sizes while viscosity and conductivity are positively correlated to increasing sizes. Moreover, voltage and flow rate appeared to be not significant for the NF sizes [27]. Moreover, the biocompatibility of PA6 had been assessed for tissue engineering scaffolds using EA.hy926 human endothelial cells [51].

In other cases, non-biodegradable polymers have been used to enhance the stability and yielding of the fiber-forming process of WK from water solutions in electrospinning, and the functional properties of the electrospun NFs, Figure 7. An example of this strategy is PEO which has been employed with WK to improve the viscometric properties of the WK solutions producing bead-free NFs [30,31] and to improve the structural properties of the resulting mats [29]. PEO is a well-known synthetic and water-soluble polymer that is easily electrospun with free-defects, Figure 7a [52,53]. Furthermore, PEO/WK-NFs have been demonstrated to keep the nanostructural form of membranes in water after thermal stabilization, Figure 7b [10,12]. In the biomedical field, the PEO/WK system has been studied in a comparative study [29]. The results showed that PEO can enhance the in vitro cell interactions altering the wettability of membranes and decreasing the hydration of WK. Since NFs of PEO/WK were obtained from water solutions, they were suitable to contain waterborne active compounds such as organic host-guest systems (Figure 7c) [14], colloids [12] and inorganic fillers (Figure 7d) [12,13]. The applications range from antibacterial [12,14] to drug delivery [13] wound dressing. Antibacterial properties of the resulting materials are excellent (85–97% bacterial reduction) against both Gram-negative and Gram-positive bacteria [12,14].

Finally, WK has been electrospun with PVP [32] obtaining bead-free NFs having an average diameter in the range of 170–290 nm. The NFs were stabilized by a thermal treatment in order to prevent dissolution during the in vitro tests. In particular, NFs treated for 18 h at 170 °C were tested following the ISO 10993-5:2009 method using primary human dermal fibroblast adult cells at passages P3–P6, and primary human epidermal keratinocyte adult cells at passages P4–P5. SEM images show that the fibroblasts were able to attach and grow on WK-based NFs [32].

## 4. WK and Biodegradable Polymers—NFs

Biodegradable polymers are commonly employed in biomedical materials such as films, sponges, hydrogels and NFs are no exception [50,54,55]. These polymers can be synthetic/natural and are degraded through biological activity—enzymatic work in the environment or in vivo [56,57], Figure 8. In environmental conditions, polymers can be decomposed by oxidation, hydrolysis, photo-degradation and biodegradation—enzymatic degradation caused by cell bioactivity, mainly bacteria and fungi [56,57]. Indeed, in the literature, the biodegradability of synthetic polymers such as PVA [58,59,60], PBS [61,62], PCL [63] and PLA [64] have been confirmed in environmental conditions. However, even so, PLA is the only one that has been extensively studied in vivo and has demonstrated hydrolysis degradation and enzymatic decomposition due to cell bioactivity [65,66,67]. It is essential to highlight that this review follows the terminology for bio-related polymers according to the IUPAC Recommendations 2012 and, by the same token, polymers abiotically degraded by enzymes in vitro are not considered biodegradable polymers [56].

Even though PVA [68], PBS [69,70] and PCL [63,71] are not biodegradable polymers in vivo, in the absence of suitable enzymes, they can undergo oxidation, hydrolysis and bioresorption over time. Furthermore, these polymers do not disrupt cell adhesion and using natural polymers such as WK increases their cell viability. From this point of view, it is important to point out the main advantages and limitations of these polymers, Table 3. Additionally, Table 4 reports the diameter distribution of WK-NFs in blend with these polymers.

From the results in Table 4, it is possible to say that the addition of WK decreases the diameter distribution of different biodegradable polymers NFs (PBS, PCL, PLA and FIB). WK probably raises the electrical conductivity of non-aqueous solutions (HFIP and FA), which boosts the bending instability and, consequently, induces the formation of NFs with smaller diameters. In contrast, the PVA case in an aqueous solution showed no variation in the diameter distribution of NFs when WK was added. However, other authors confirmed the effect of adding WK when FA was employed as a non-aqueous solvent for the same blend of PVA and WK [74]. All these results show the relevance of using WK in biodegradable polymer blends.
jfb-14-00005-t004_Table 4Table 4Morphological features of WK-NFs with biodegradable polymers.Polymer Blend (% wt.)Polymer Mw * (kDa)SolventMax. Freq. Diameter (nm)Minimum Diameter (nm)Maximum Diameter (nm)Range (μm)ReferencesPVA130Water45011014601.35[33]17/83 WK/PVA130Water45011014601.35[33]50/50 WK/PVA75FA75351250.09[74]PBS50HFIP56312013001.18[35]50/50 WK/PBS50HFIP23629020206508100.630.79[34][35]PCL65HFIP1711002500.15[36]50/50 WK/PCL65HFIP12414430502102600.180.21[36][37]PLA138HFIP6004008000.40[75]50/50 WK/PLA119HFIP120502000.15[15]FIBn.a.FA8007016201.55[38]50/50 WK/FIBn.a.FA220504000.35[38]GELn.a.FA114303000.27[39]23/77 WK/GELn.a.FA109303000.27[39]19/19/62 WK/SER/GELn.a.FA2003015001.47[39]* Mw is not available for the second natural polymer (FIB, GEL and SER).


Regarding the use of natural biodegradable polymers, the addition of WK in non-aqueous solutions has a positive, negligible and negative effect on the diameter distribution of NFs made of FIB, GEL and SER, respectively [39]. A positive effect was observed when FIB was used, the considerable reduction in diameter distribution means that WK contains more amino-acid side chains protonated than FIB. In WK/SER/GEL case, the negative effect on NFs diameters can be caused by an unstable electrospinning solution with multiple components.

The literature has pointed out that WK not only decreases the diameter distribution of polymeric NFs, but also improves the hydrophilicity of NFs made of protein and synthetic polymers, particularly for PCL and WK [36], Figure 9. Then, the presence of WK inside NFs scaffolds gives rise to better cell material interactions during in vitro tests and increases the proliferation of hMSCs [36]. Therefore, the wettability and the stability of membranes in aqueous solutions are key properties that must be considered for biomedical materials. In the literature, it has also been reported that NFs of pure synthetic polymers (such as PBS [76], PCL [36] and PLA [73]) produce hydrophobic membranes with water contact angles (WCAs) over 100°. Meanwhile, the NFs with proteins such as WK or GEL have been demonstrated to be hydrophilic with WCAs below 50° [36,73,77]. These results present evidence of the positive contribution of adding natural biopolymers such as WK, whose macromolecular structure contains a great deal of hydrophilic and polar groups in its structure. The WCA of NFs can also be modified using water-soluble polymers such as PVA, whose side hydroxyl groups form strong hydrogen bonds and stabilize the α−helix conformations of proteins [33].

As far as the solvent effect is concerned, it is necessary to consider that FA reports a greater dielectric constant value than HFIP, which facilitates the production of NFs with smaller diameter distribution [10,78]. Table 4 shows that the smallest NFs were obtained when FA was used instead of HFIP, these results are in agreement with those from pure WK-NFs in the same solvents [10]. Despite the fact that water has a greater dielectric constant than HFIP and FA, up to now it was not possible to obtain pure WK-NFs by using only water as a solvent. Furthermore, FA and HFIP share a similar range of acidities [79], so both similarly protonate the amino-acid side chains of WK and increase the electrical conductivity of non-aqueous solutions during electrospinning. As mentioned above, this is the main reason for smaller diameter distribution in non-aqueous solvents compared to water. Additionally, it is important to say that HFIP is a great hydrogen bond donor [10]. This property can stabilize α–helix conformations within WK-NFs and these scaffolds have better cell viability than membranes obtained from FA [10]. Indeed, PVA has demonstrated the same capacity of forming a lot of hydrogen bonds and increasing the content of α–helix structures within WK-NFs [33]. To sum up some solvent effects on protein-NFs, a schematic summary of all main features able to influence nanofiber formation were reported in Figure 10. Noteworthy, other parameters such as surface tension, volatility (vapor pressure) and viscosity, that depend on those reported in the scheme, and they play a relevant role on the microscopic phenomena are involved in the fiber formation during the electrospinning.

Finally, the crystallinity of NFs is an important parameter in the biodegradation of polymers, as the crystalline regions of polymers hinder the degradation of polymeric scaffolds contrary to the amorphous regions [57]. PBS [69], PCL [63], PLA [65] and PVA [72] are semi-crystalline polymers whose crystallinity in NFs-membranes can be reduced by the addition of WK. Additionally, solvents such as HFIP not only rise the content of α−helix structures but can also increase the crystallinity of nanofibrous membranes as a consequence of a greater dielectric constant value [78]. It is also essential to consider the interaction between protein and synthetic polymer. For example, PVA and WK have simultaneously demonstrated an increase in α−helix conformation and PVA-crystallinity due to new hydrogen bonds between hydroxyl groups of PVA and amino acid side chains of WK [33]. Furthermore, when these PVA/WK-NFs were treated at 180 °C for 2 h, a reduction in PVA-crystallinity was observed, which enhanced the cell adhesion and cell viability of hMSCs [33]. Likewise, in other experiments, the addition of WK in GEL-NFs induced a reduction in GEL-crystallinity and further addition of WK+SER in GEL-NFs caused smaller values of GEL-crystallinity than in the previous case [39]. The synergistic effect of adding KER+SER in GEL-NFs not only decreased GEL-crystallinity but also improved the proliferation of hMSCs over time during in vitro tests [39]. From these results, it can be said that NFs with smaller crystallinity demonstrated a better cell–material interaction, improved biodegradability and a greater proliferation of hMSCs.

## 5. Conclusions

WK is a natural biodegradable polymer that can be electrospun to produce pure/composite NFs from aqueous or non-aqueous solutions. By and large, it is necessary to integrate this fibrous protein with synthetic and biocompatible polymers. The solvent selection varies according to the polymer blend affecting properties such as the morphology of NFs, wettability, biodegradability, crystallinity and cell–material interaction. In fact, the use of FA forms NFs with the smallest diameter distribution from pure-WK and WK/biodegradable polymers (PLA, FIB and GEL). Nevertheless, WK-NFs with a greater diameter distribution from solvents such as HFIP offers the possibility of employing polymers (PBS and PCL) with better mechanical properties and improved water stability. When an aqueous solution is required during the electrospinning process, a water-soluble polymer such as PVA can be used so that WK-NFs could have more wettability. It is worth mentioning that these polymer blends could need post-treatments to enhance the water stability or mechanical properties of WK-scaffolds which affects the crystallinity and biodegradability of NFs.

To summarize, the application of this structural protein is varied thanks to the multiple options of solvent and polymer blends used in the production of nanofibrous membranes.

## Figures and Tables

**Figure 1 jfb-14-00005-f001:**
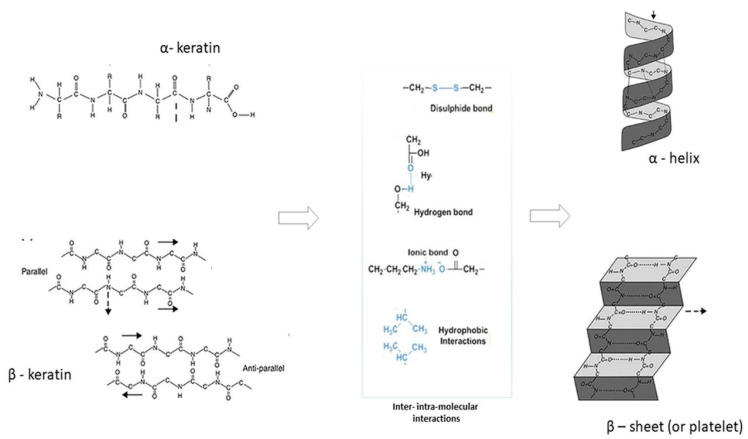
Chemical configuration of α- and β- keratin, inter and intra-molecular interactions and secondary structures (α-helix and β-sheet) [3] (with copyright permission).

**Figure 2 jfb-14-00005-f002:**
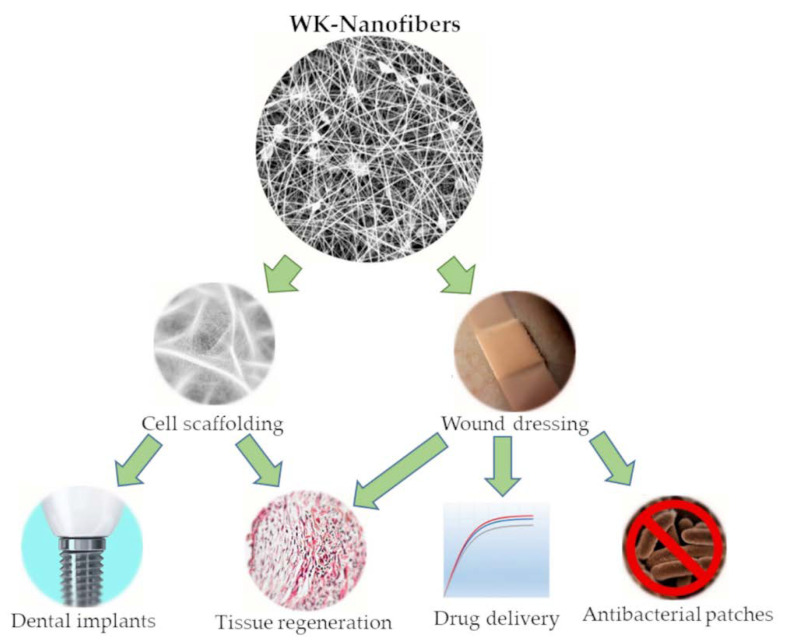
Biomedical applications of WK-NFs.

**Figure 3 jfb-14-00005-f003:**
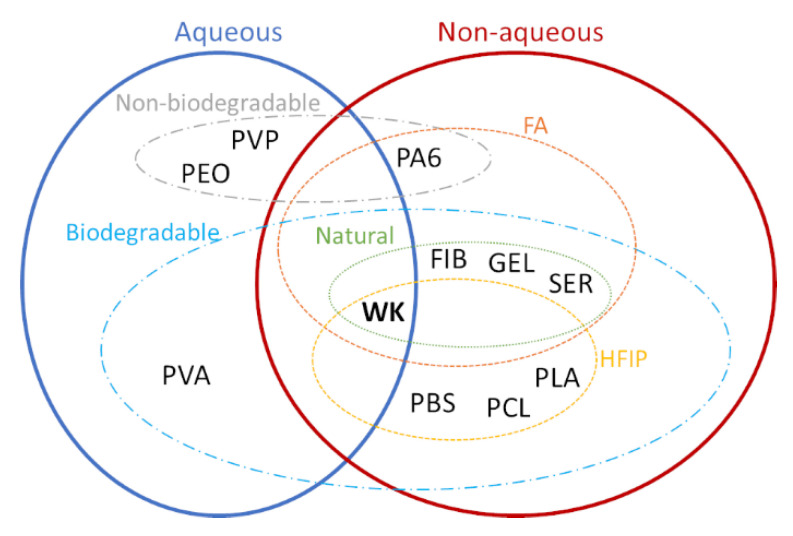
Solvents and polymers used during the electrospinning of WK.

**Figure 4 jfb-14-00005-f004:**
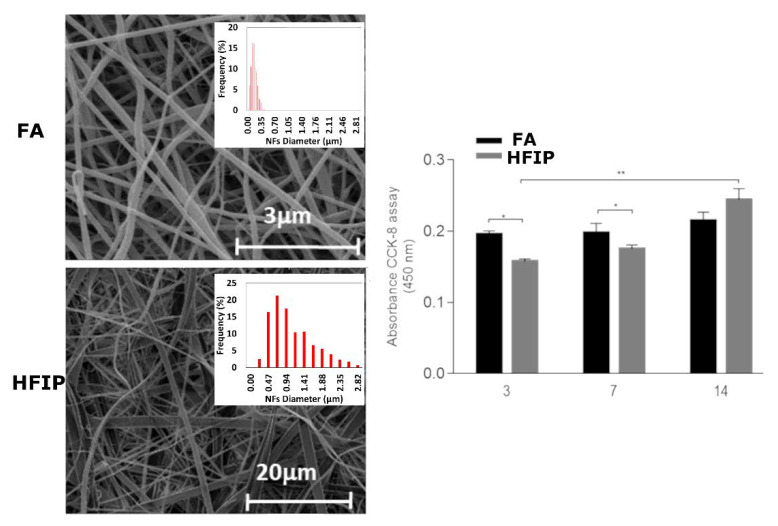
Comparison of WK fiber morphology produced by the dissolution in different solvents: FA and HFIP. Morphology and in vitro studies (* *p* < 0.05, ** *p* < 0.01). (adapted from [10]).

**Figure 5 jfb-14-00005-f005:**
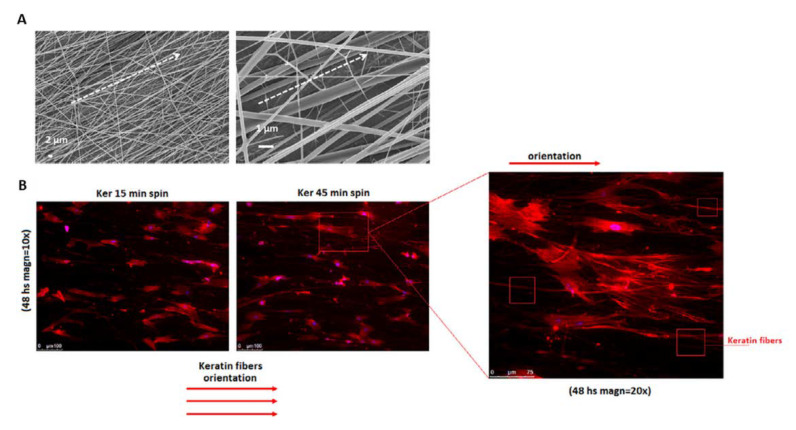
WK-NFs with preferential alignment onto Titanium plugs for dental restoration: (**A**) fiber morphology and (**B**) in vitro spreading and growth of fibroblasts. Adapted from [11].

**Figure 6 jfb-14-00005-f006:**
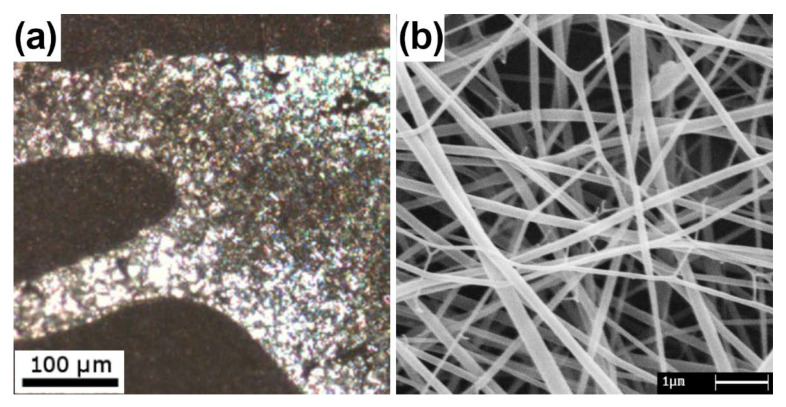
Phase segregation in a WK/PA6 70/30 blend film observed by light microscopy (**a**) and WK/PA6 70/30 electrospun NFs observed by SEM (**b**). Unpublished data from [27].

**Figure 7 jfb-14-00005-f007:**
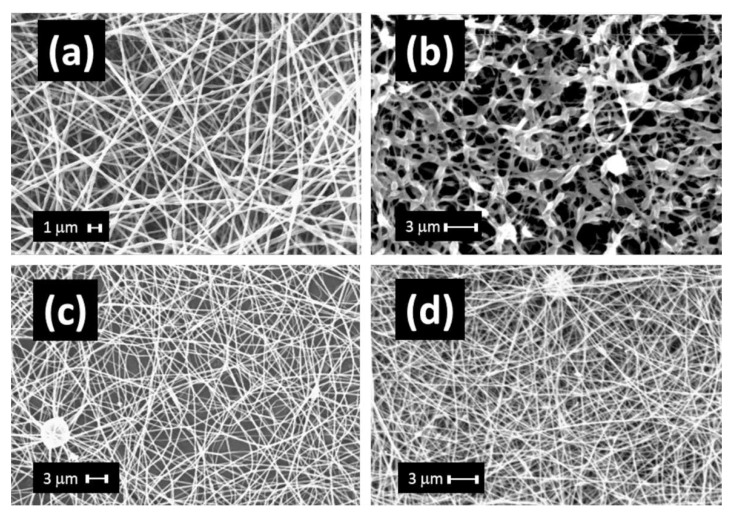
WK-NFs using PEO: (**a**) pure PEO-NFs, (**b**) WK/PEO-NFs after 180 °C for 2 h and 18 h in water, (**c**) WK/PEO-NFs with bili acid binding protein and irgasan, (**d**) WK/PEO-NFs with surfactant and Ag-Nanoparticles (average diameter ≤ 217 nm). Unpublished data from [12,14,53].

**Figure 8 jfb-14-00005-f008:**
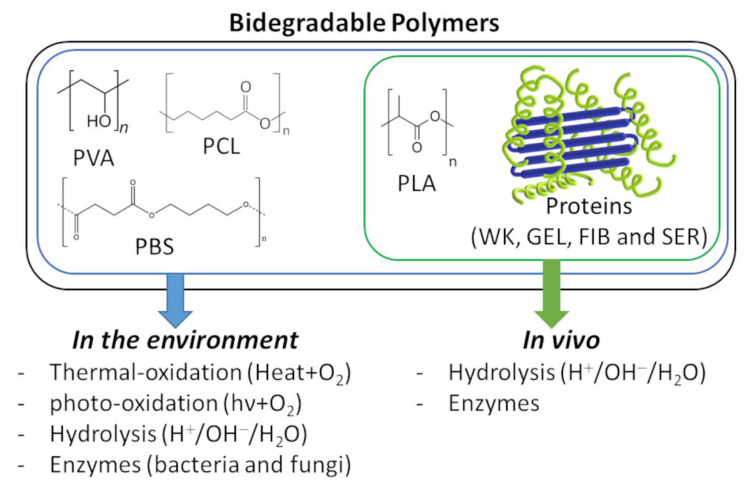
Biodegradable polymers employed in WK-NFS and their biodegradability in vivo and in the environment.

**Figure 9 jfb-14-00005-f009:**
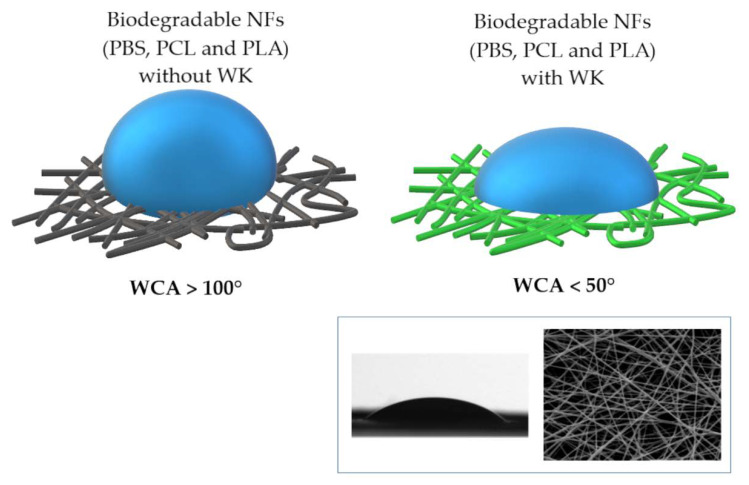
Effect of WK on NFs wettability. PCL/WK-NFs (in the same square): SEM image of fiber morphology and WCA at zero time (ca. 44°). Adapted from [34].

**Figure 10 jfb-14-00005-f010:**
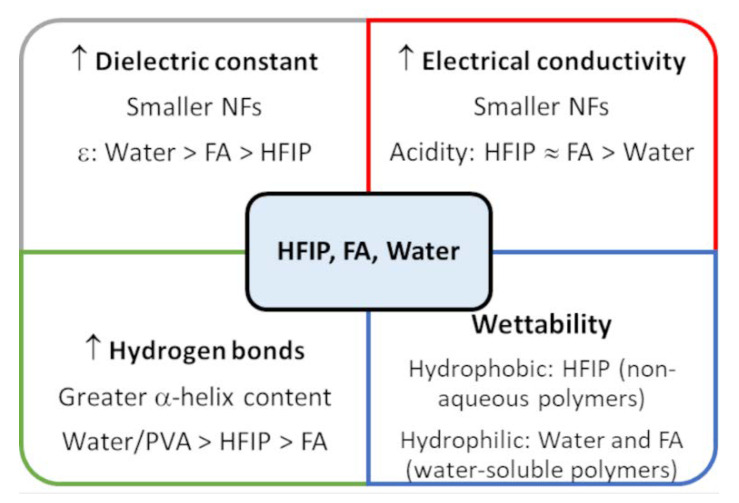
Summary of physical properties (i.e., dielectric constant, electrical conductivity, hydrogen bonding and wettability) of solvents used for the fabrication of WK-NFs. “↑” means increase.

**Table 1 jfb-14-00005-t001:** Pure and blend—WK-NFs.

Polymer Blend (% wt.)	Polymer Mw * (kDa)	Solvent	Biodegradability	Biocompatibility	Reference(s)
100 WK	n.a.	FA	+	+	[4,9,26]
100 WK	n.a.	HFIP	+	+	[10,11]
50/50 WK/PA6	22	FA	−	+	[27,28]
70/30 WK/PEO	400	Water	−	+	[29,30,31]
25/75 WK/PVP	1300	Water	−	+	[32]
17/83 WK/PVA	130	Water	+	+	[33]
50/50 WK/PBS	50	HFIP	+	+	[34,35]
50/50 WK/PCL	65	HFIP	+	+	[36,37]
50/50 WK/PLA	119	HFIP	+	+	[15]
50/50 WK/FIB	n.a.	FA	+	+	[38]
23/77 WK/GEL	n.a.	FA	+	+	[39]
19/19/62 WK/SER/GEL	n.a.	FA	+	+	[39]

* Mw is not available for the second polymer (FIB, GEL and SER). WK: wool keratin, PA6: polyamide 6, PEO: polyethylene oxide, PVP: polyvinylpyrrolidone, PVA: polyvinyl alcohol, PBS: polybutylene succinate, PCL: polycaprolatone, PLA: polylactic acid, FIB: fibroin, GEL: gelatin, SER: sericin, FA: Formic acid and HFIP: hexafluoroisopropanol. “+” means biodegradable/biocompatible; “−” means non biodegradable; “n.a.” means not applicable.

**Table 2 jfb-14-00005-t002:** Main electrospinning conditions used in the production of WK-NFs.

NFs	Additional Polymer(s)	Solvent System	Overall Polymer Conc.	WK Conc. (% on Polymer)	Voltage (kV)	Flow Rate (mL min^−1^)	Working Distance (cm)	Needle I.D. or Gauge	Collector	Reference(s)
WK	n.a.	FA	15% *w*/*w*	100	25	0.003	15	0.2 mm	Flat square	[5,9,26]
WK	n.a.	HFIP	10% *w*/*v*	100	25	0.008	15	18 Ga	Rotating drum	[11]
WK/PA6	PA6, 22 kDa	FA	15% *w*/*w*	0–100	15, 20, 25, 30	0.001, 0.005, 0.01	15	0.4 mm	Rotating flat disk	[27]
WK/PEO	PEO, 400 kDa	Water	7% *w*/*w*	10–90	20	0.01	20	0.2 mm	Rotating flat disk	[29]
WK/PEO	PEO, 400 kDa	Water	5, 7, 10% *w*/*w*	50	10–30	0.01–0.03	20	-	Rotating flat disk	[31]
WK/PVP	PVP, 1300 kDa	Water	7.6% *w*/*v*	25	18	0.01	20	18 Ga	Flat disk	[32]
WK/PVA	130 kDa	Water	8.8, 10% *w*/*w*	17, 33	25	0.015	25, 30	0.4 mm	Flat plate	[33]
WK/PBS	PBS, 50 kDa	HFIP	13% *w/v*	30, 50, 70	18	0.03	18	-	Flat square	[34]
WK/PBS	PBS, 50 kDa	HFIP	15% *w*/*v*	50	20	0.03	15	0.8 mm	Flat square	[35]
WK/PCL	PCL, 65 kDa	HFIP	10% *w*/*v*	50, 70	15–25	0.0017	9–12	-	-	[37]
WK/PLA/GO	PLA, 119 kDa	HFIP	10% *w*/*v*	50	12, 15, 18	0.03	12, 15, 18	0.603 mm	Flat plate	[15]
WK/FIB	FIB	FA	15% *w*/*w*	0–100	30	0.005	10	0.2 mm	Rotating flat disk	[38]
WK/GEL	GEL	FA	13% *w*/*w*	23	30–35	NFE	13	NFE	Flat PP-NW	[39]
WK/GEL/SER	GEL, SER	FA	16% *w*/*w*	19	30–35	NFE	13	NFE	Flat PP-NW	[39]

GO: graphene oxide, Conc.: concentration, Ga: gauge, NFE: needle-free electrospinning, PP-NW: polypropylene non-woven.

**Table 3 jfb-14-00005-t003:** Advantages and limitations of biodegradable polymers used in WK-NFs.

Biodegradable Polymer	Advantages	Limitations	Reference(s)
PVA	- Its hydroxyl groups can create physical/chemical interactions with other molecules. - It is a hydrophilic and water-soluble polymer. - It has chemical stability and transport properties that characterize ionotropic polymer. - It has good transparency, good mechanical and thermal properties. - It is resistant to oxygen permeation. - It has a wide range of applications in different industrial–commercial segments.	- Its processing requires relatively large amounts of water and organic plasticizers in extrusion processes. - It can be efficiently degraded by microorganisms whose occurrence in natural environments may be relatively uncommon. - Pure PVA films have disadvantages such as brittleness, low fracture elongation, poor water resistance and processability, which limit its wide application.	[33,58,60,72]
PBS	- Its good heat resistance and melting temperature provide a wide processing range. - It has good thermal stability and excellent mechanical properties (comparable to polyethylene and polypropylene). - It contributes to improving the mechanical properties of polymer blend NFs. - It has good clarity, great processability and flexibility. - Its ester bonds can be hydrolyzed by water.	- It is slightly brittle. - Its slow crystallization rate, low melt viscosity and softness limit PBS processing and applications, especially in injection molding. - The strength properties of PBS deteriorate due to a rapid crystallization reaction when combined with other materials. - It has insufficient osteoblast compatibility and bioactivity.	[34,61,70]
PCL	- It is easy to manufacture and manipulate into an extensive range of implants and devices thanks to its rheological and viscoelastic properties. - Its crystallinity tends to decrease by increasing its molecular weight. - It has multiple potential applications in the biomedical field thanks to its low melting point (59–64 °C)	- It can be only degraded by outdoor living organisms (bacteria and fungi). - Its bioresorbability process takes much longer, first propagating through hydrolytic degradation. - Pure PCL scaffolds cannot trigger cell adhesion mechanisms due to their intrinsic hydrophobic properties.	[36,63]
PLA	- It is widely used for biomedical scaffolds and implants with theranostics and drug delivery systems. - It is simple to synthesize and can be tailored for different therapeutic applications. - It is naturally degraded over time into well-tolerated and safe degradation products, which are secreted from the body. - It has good biocompatibility and mechanical properties.	- It is highly hydrophobic and lacks a cell recognition site, which hinders the rapid adhesion, migration and regeneration of tissue cells.	[65,73]

## Data Availability

Not applicable.

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
