# Peer review of "Wool Keratin Nanofibers for Bioinspired and Sustainable Use in Biomedical Field"

_jfb, 2022, doi:10.3390/jfb14010005_

Round 1
Reviewer 1 Report
Well-written and well-structured article. The title matches with the content. It is of interest in the area. The article is a review that, from my point of view, will serve as a reference for work in keratin and electrospinning. The information regarding the effect of solvents on the thickness of the nanofilament is very important.
I have only one comment: define the acronyms of each table at the bottom. For example: WK: Wool Keratin; FA: Formic Acid; ...
Author Response
Well-written and well-structured article. The title matches with the content. It is of interest in the area. The article is a review that, from my point of view, will serve as a reference for work in keratin and electrospinning. The information regarding the effect of solvents on the thickness of the nanofilament is very important.
I have only one comment: define the acronyms of each table at the bottom. For example: WK: Wool Keratin; FA: Formic Acid; ...
R:/ As requested, the acronyms have been defined in Table 1.
Reviewer 2 Report
J. Funct. Biomater.
In this manuscript, the authors provide a comprehensive study regarding the Design of Wool keratin nanofibers for bioinspired and sustainable use in the biomedical field. It is eye-catching and exciting to a researcher in this field. However, some corrections and explanations must be done previous to publish the paper in the J. Funct. Biomater. Some of my comments and questions on this manuscript are as follows:
- The manuscript contains huge textual documentation, which makes it very boring for the readers. I would suggest the authors add some of the figures or add new images to the single image, such as Figure 4, Figure 6, and Figure 7. and schematic illustrations to make it more interesting.
- The caption of Figure 6, and Figure 7 should be modified, and it should be illustrated more comprehensively.
- I strongly suggest that the author presents the schematic presentation of the essential application of wool keratin nanofibers employed for biomedical and tissue engineering fields, including wound dressing.
- The advantages and limitations of biodegradable polymers such as PVA, PBS, PCL, and PLA in environmental conditions should be stated.
- The author should provide information about the electrospinning conditions of wool keratin nanofibers in the form of a Table.
- The schematic illustration regarding wool keratin hydrogel nanofibers, including disulfide bonds, hydrophobic interactions, and hydrogen bonds could be presented to benefit readers.
- The author stated "One of the main drawbacks ..........also improving stiffness and mechanical strength [35]". However, there is no report was found regarding mechanical properties of Wool keratin nanofibers for biomedical and tissue engineering applications.
8. Some references about natural and synthetic nanofibers polymers and polymers may be useful for this Review Article: Materials 2020, 13, 2153; Adv. Eng. Mater . 2022, 24, 2101460; Carbohydrate Polymers 291 (2022) 119670.
- Surprisingly small reference to the J. Funct. Biomater in the literature despite the large relevant literature there. This should be improved. There are several important papers in the recent literature.
Author Response
In this manuscript, the authors provide a comprehensive study regarding the Design of Wool keratin nanofibers for bioinspired and sustainable use in the biomedical field. It is eye-catching and exciting to a researcher in this field. However, some corrections and explanations must be done previous to publish the paper in the J. Funct. Biomater. Some of my comments and questions on this manuscript are as follows:
- The manuscript contains huge textual documentation, which makes it very boring for the readers. I would suggest the authors add some of the figures or add new images to the single image, such as Figure 4, Figure 6, and Figure 7. and schematic illustrations to make it more interesting.
R:/ Figure 6 has been corrected and changed to Figure 8. At the same time, Figure 7 has been replaced with Figure 9 and it has been integrated with other images and data. Additionally, two new figures have been added (Figure 2 and Figure 1).
- The caption of Figure 6, and Figure 7 should be modified, and it should be illustrated more comprehensively.
R:/ As requested, the figures' captions have been modified.
- I strongly suggest that the author presents the schematic presentation of the essential application of wool keratin nanofibers employed for biomedical and tissue engineering fields, including wound dressing.
R:/ Figure 2 has been added to present the biomedical applications of wool keratin nanofibers.
- The advantages and limitations of biodegradable polymers such as PVA, PBS, PCL, and PLA in environmental conditions should be stated.
R:/ Table 3 has been included to highlight the Advantages and limitations of biodegradable polymers used in wool keratin nanofibers
- The author should provide information about the electrospinning conditions of wool keratin nanofibers in the form of a Table.
R:/ Table 2 has been added to provide information about the main electrospinning conditions used in the production of wool keratin nanofibers.
- The schematic illustration regarding wool keratin hydrogel nanofibers, including disulfide bonds, hydrophobic interactions, and hydrogen bonds could be presented to benefit readers.
R:/ Thank you for the comment. A new Figure (Figure 1) has been included in order to schematically describe the chemical structure of keratins as a function of weak chemical bonds and interactions. For this figure, copyright permissions have been requested.
- The author stated "One of the main drawbacks ..........also improving stiffness and mechanical strength [35]". However, there is no report was found regarding mechanical properties of Wool keratin nanofibers for biomedical and tissue engineering applications.
R:/ It has been corrected as follows:
“One of the main drawbacks in the fabrication of biologically stable substrates is related to the conditions used during the extraction process, which may drastically influence the degradation mechanisms of the protein. … Hence, a strict control of these processes may ensure the reproduction of a-helix-like structure that is typical of other biological proteins such as collagen, also improving stiffness and mechanical strength [40].”
- Some references about natural and synthetic nanofibers polymers and polymers may be useful for this Review Article: Materials 2020, 13, 2153; Adv. Eng. Mater. 2022, 24, 2101460; Carbohydrate Polymers 291 (2022) 119670.
R:/ Thank you for this comment, the suggested references have been cited in the text in some pertinent points.
- Surprisingly small reference to the J. Funct. Biomater in the literature despite the large relevant literature there. This should be improved. There are several important papers in the recent literature.
R:/ Seven new references from J. Funct. Biomater. have been included in the introduction as follows:
“Finally, it is essential to remember that the electrospinning process can be extended to many other polymers (i.e. collagen, chitosan, polycaprolactone and polyhydroxy butyrate [16–18]) to obtain nanofibers with drug delivery systems [19] and anti-microbial properties [20] for therapeutic use [21] and wound healing applications [22].”
- Khodir, W.K.W.A.; Razak, A.H.A.; Ng, M.H.; Guarino, V.; Susanti, D. Encapsulation and Characterization of Gentamicin Sulfate in the Collagen Added Electrospun Nanofibers for Skin Regeneration. J. Funct. Biomater. 2018, Vol. 9, Page 36 2018, 9, 36, doi:10.3390/JFB9020036.
- Zhou, Y.; Li, Y.; Li, D.; Yin, Y.; Zhou, F.L. Electrospun PHB/Chitosan Composite Fibrous Membrane and Its Degradation Behaviours in Different pH Conditions. J. Funct. Biomater. 2022, Vol. 13, Page 58 2022, 13, 58, doi:10.3390/JFB13020058.
- Rahman, S.; Carter, P.; Bhattarai, N.; Puoci, F. Aloe Vera for Tissue Engineering Applications. J. Funct. Biomater. 2017, Vol. 8, Page 6 2017, 8, 6, doi:10.3390/JFB8010006.
- Wang, Y.; Yu, D.-G.; Liu, Y.; Liu, Y.-N.; Wang, Y.; Yu, D.-G.; Liu, Y.; Liu, Y.-N. Progress of Electrospun Nanofibrous Carriers for Modifications to Drug Release Profiles. J. Funct. Biomater. 2022, Vol. 13, Page 289 2022, 13, 289, doi:10.3390/JFB13040289.
- Hamdan, N.; Yamin, A.; Hamid, S.A.; Khodir, W.K.W.A.; Guarino, V. Functionalized Antimicrobial Nanofibers: Design Criteria and Recent Advances. J. Funct. Biomater. 2021, Vol. 12, Page 59 2021, 12, 59, doi:10.3390/JFB12040059.
- Papa, A.; Guarino, V.; Cirillo, V.; Oliviero, O.; Ambrosio, L. Optimization of Bicomponent Electrospun Fibers for Therapeutic Use: Post-Treatments to Improve Chemical and Biological Stability. J. Funct. Biomater. 2017, Vol. 8, Page 47 2017, 8, 47, doi:10.3390/JFB8040047.
- Azimi, B.; Maleki, H.; Zavagna, L.; de la Ossa, J.G.; Linari, S.; Lazzeri, A.; Danti, S. Bio-Based Electrospun Fibers for Wound Healing. J. Funct. Biomater. 2020, Vol. 11, Page 67 2020, 11, 67, doi:10.3390/JFB11030067.
Reviewer 3 Report
Keratins is a family of wide-spread proteins as the components of animal wool and hair fibers. It is a rather interesting protein family in natural physiology and the source for different biotechnological and medical applications. Thus, every new information on the structure and properties of these proteins are useful.
Unfortunately, I cannot recommend the reviewed article “Design of wool keratin nanofibers for bioinspired and sustainable use in biomedical field” for publication, because its construction and the form of knowledge presentation make it non-informative for the reader. It is very difficult to find clear logic and the cause-and-effect relations between protein molecular structure, micromorphology and properties of studied keratins. I didn’t find clear examples of keratin use in the declared biomedical field and cannot understand the contribution of Italian scientists in the studied field. The only obvious deposit is too large percentage of self-citing – about one third of citing papers.
Besides there is very obscure information in some places in the text. For example, I cannot understand in Introduction what is “cysteine-rich amino acids”, how the “covalent bond is responsible for preventing the dissolution of fibers in water”, how poor solubility of keratin provides for its good biocompatibility and biodegradability, etc.?
Also, the English of this article is not good.

Author Response
Keratins is a family of wide-spread proteins as the components of animal wool and hair fibers. It is a rather interesting protein family in natural physiology and the source for different biotechnological and medical applications. Thus, every new information on the structure and properties of these proteins are useful.
Unfortunately, I cannot recommend the reviewed article “Design of wool keratin nanofibers for bioinspired and sustainable use in biomedical field” for publication, because its construction and the form of knowledge presentation make it non-informative for the reader.
R:/ As reported in the abstract, this work is a Review rather than an “Original research manuscript” that is focused on the latest progress made in wool keratin nanofibers for the biomedical field. Therefore, this review presents and provides updates regarding the production and use of wool keratin nanofibers in the biomedical field. Nevertheless, the title has been changed to “Wool keratin nanofibers for bioinspired and sustainable use in the biomedical field” in order to prevent readers from confusing the aim of this work.
Abstract: “Keratin nanofibers have received significant attention in biomedical applications, such as tissue engineering and cell growth scaffolds, for their biocompatibility and bio-functionality. Accordingly, it is proposed an extensive overview of recent studies focused on the optimization of keratin based nanofibers,…”
It is very difficult to find clear logic and the cause-and-effect relations between protein molecular structure, micromorphology and properties of studied keratins.
R:/ To begin with, the introduction reported how this review would be sorted out according to the production of wool keratin nanofibers and their blends.
Introduction: “As evidenced by Figure 3 and Table 2, WK is a versatile protein that can be electrospun under different conditions in water, non-aqueous solvents and offers a wide range of options for polymer blends. Despite the fact that all those polymers are biocompatible, not all of them are biodegradable. Therefore, this review will discuss the morphology, the stability of NFs in water, the selection of solvent, the response of WK-NFs during in-vitro test and the potential application of these functional biomaterials, considering mainly three groups: pure WK-NFs, NFs made of non-biodegradable polymers with WK and NFs made of biodegradable polymers with WK.”
On the other hand, this work is a review and it was not submitted to be an “Original research manuscript” about the cause-and-effect relations between protein molecular structure and keratin properties. Indeed, this review gives an overview of the latest publications carried out in Italy considering the main properties of wool keratin nanofibers which is in agreement with the aim of this work.
I didn’t find clear examples of keratin use in the declared biomedical field and cannot understand the contribution of Italian scientists in the studied field. The only obvious deposit is too large percentage of self-citing – about one third of citing papers.
R:/ The common biomedical applications of wool keratin nanofibers have been pointed out in Figure 2. Furthermore, the introduction has been modified as follows:
“In order to exploit the natural properties of wool keratin (WK) as a structural protein, after extraction and purification, WK can be processed to produce scaffolds – interconnected structures similar to Extra Cellular Matrix (ECM). This goal can be reached by electrospinning, a technique that generates nanofibrous systems which can mimic the structural environment of ECM and support the growth of cells in applications like tissue engineering [9,10] and dental implants [5,11]. In wound dressing, WK-nanofibers could also have application in the production of nanofibers membranes with antibacterial properties [12,13] and drug delivery systems [14,15]. The common applications of WK-nanofibers have been presented in Figure 2.”
As far as the contribution of Italian scientists in this field is concerned, a paragraph had been added in the introduction to highlight the contribution of Italian researchers on the production and application of wool keratin nanofibers in comparison with other countries.
Introduction: “It is worth mentioning that the research on nanofibers (NFs) production by electrospinning has been growing worldwide over time. According to the amount of publications in Scopus and Web of Science, China, United States, South Korea, Iran, India, Germany, Japan, United Kingdom, Turkey and Italy currently report the major number of publications in this field. Among them, Italy and China have published the most significant number of articles on the production and application of WK-NFs. As part of the special issue State-of-the-Art Functional Biomaterials in Italy, this review will deal with scientific publications that have been exclusively conducted in Italy.”
Finally, the paper has been proposed for the Special Issue "State-of-the-Art Functional Biomaterials in Italy" (https://www.mdpi.com/journal/jfb/special_issues/func_biomat_italy).
Besides there is very obscure information in some places in the text. For example, I cannot understand in Introduction:
what is “cysteine-rich amino acids”
R:/ “cysteine-rich amino acids” has been changed to “cysteine-rich proteins..”.
how the “covalent bond is responsible for preventing the dissolution of fibers in water”.
R:/ Keratin is a no water-soluble protein before extraction due to the fact H2O molecules are not capable of breaking disulfide bonds. Keratin is commonly extracted from wool fibers or any other natural source by oxidation, reduction or sulfitolysis. In order to increase the extraction yield of keratin, urea is usually used as a denaturing agent of hydrogen bonds, as well. Nevertheless, urea cannot break disulfide bonds either. As a result, the disulfide covalent bond has to be broken by oxidation, reduction or sulfitolysis, keratin cannot be extracted and separated from fibers in aqueous solutions. If the Reviewer wants to deepen into this information, the authors could suggest some references (i.e., https://doi.org/10.1007/978-3-030-02901-2). However, the extraction of keratin has been extensively discussed in the literature and the aim of work is not keratin extraction. As disulfide bonds were broken, the keratin became water soluble.
how poor solubility of keratin provides for its good biocompatibility and biodegradability, etc.?
After extraction, keratin is a water-soluble protein processed to obtain different biomaterials and, successively, those are crosslinked to prevent the dissolution of keratin-based biomaterials in water. In particular, this review discussed the production of wool keratin nanofibers. As reported in the introduction, the biocompatibility and biodegradability of wool keratin nanofibers have already been reported in the literature and multiple references had been added.
Also, the English of this article is not good.
R:/ The English has been checked and corrections have been made where required.
Round 2
Reviewer 3 Report
After corrections I can recommend publishing article in the present form.